# Use of State Sequence Analysis in Pharmacoepidemiology: A Tutorial

**DOI:** 10.3390/ijerph182413398

**Published:** 2021-12-20

**Authors:** Jacopo Vanoli, Consuelo Rubina Nava, Chiara Airoldi, Andrealuna Ucciero, Virginio Salvi, Francesco Barone-Adesi

**Affiliations:** 1London School of Hygiene and Tropical Medicine (LSHTM), London WC1E 7HT, UK; jacopo.vanoli@lshtm.ac.uk; 2School of Tropical Medicine and Global Health (TMGH), Nagasaki University, Nagasaki 852-8521, Japan; 3Department of Economics and Statistics “Cognetti de Martiis”, University of Turin, 10124 Turin, Italy; 4Department of Translational Medicine, University of Eastern Piedmont, 28100 Novara, Italy; chiara.airoldi@uniupo.it (C.A.); andrealuna.ucciero@uniupo.it (A.U.); 5Department of Neuroscience, ASST Fatebenefratelli Sacco, 20157 Milan, Italy; virginiosalvi@gmail.com (V.S.); francesco.baroneadesi@uniupo.it (F.B.-A.)

**Keywords:** state-sequence analysis, pharmacoepidemiology, data-mining

## Abstract

While state sequence analysis (SSA) has been long used in social sciences, its use in pharmacoepidemiology is still in its infancy. Indeed, this technique is relatively easy to use, and its intrinsic visual nature may help investigators to untangle the latent information within prescription data, facilitating the individuation of specific patterns and possible inappropriate use of medications. In this paper, we provide an educational primer of the most important learning concepts and methods of SSA, including measurement of dissimilarities between sequences, the application of clustering methods to identify sequence patterns, the use of complexity measures for sequence patterns, the graphical visualization of sequences, and the use of SSA in predictive models. As a worked example, we present an application of SSA to opioid prescription patterns in patients with non-cancer pain, using real-world data from Italy. We show how SSA allows the identification of patterns in prescriptions in these data that might not be evident using standard statistical approaches and how these patterns are associated with future discontinuation of opioid therapy.

## 1. Introduction

In the last decade, enormous progress has been made in the use of real-world data to provide information on drug use, effectiveness and safety. This has been principally due to the expansion of information technology with easier access to diverse sources such as electronic health records (EHR), administrative or health claims data, as well as disease and drug monitoring registries [1]. There is thus a pervasive need to adapt research designs and statistical methods to such rapid evolution, which affects the amount, the frequency, the type, and the nature of available information [2]. In this framework, classical statistical tools can be integrated with novel data mining techniques suitable of identifying patterns in complex data structures. Thus far, in pharmacoepidemiology, these techniques have been mainly used to identify adverse drug reactions [3], while their application to the analysis of drug prescriptions to identify longitudinal use patterns is still limited [4]. Indeed, most drug utilization studies currently rely on simple adherence measures such as medication possession ratio and proportion of days covered [5]. While these summary indicators have different advantages, they typically fail to identify the different prescription patterns of patients, which in turn can be associated with important health-outcomes [6]. To fill this gap, in this paper, we show the application of state sequence analysis (SSA) to pharmacoepidemiological data to evaluate the temporal order of prescriptions and to identify latent complex patterns. Without assuming a priori hypothesis, SSA is an effective tool to study distinctive features of homogeneous groups of sequences, exploiting their pairwise dissimilarities in unsupervised clustering [7]. The SSA method indeed looks at the life sequence as a single unit of analysis, extracts fundamental descriptive information, and makes the data easier to comprehend, differently from event history analysis [8]. Identifying in sequences also typical trajectories and recurring structures, the SSA is considered the most suitable tool for signal detection in healthcare databases [3].

While SSA has been long used in social sciences, especially for labor and family mobility research [9,10,11,12,13], its use in pharmacoepidemiology is still in its infancy [14,15,16]. SSA allows a simple and compact representation of life courses identical to the one used to code DNA molecules [9]. Thus, in social sciences, SSA is employed, for instance, to model demographic projections based on microsimulation methods [9], family file events [12], early employment insecurity [13], pathways to adulthood [11], and the prevalence of nuclear families [10]. Moreover, SSA in pharmacoepidemiology proves to be effective in evaluating the impact of regulatory measures on the prescription of sedative medications [15]; describing longitudinal patterns of disease-modifying therapies usage, grouping the population [16]; and investigating the conformity of prescribing practices of respiratory drug treatments [14].

As a worked example, we present here an application of SSA to opioid prescription patterns in patients with non-cancer pain, using real-world data from Italy. The choice of this example is partly motivated by the fact that, while the use of opioids for the management of chronic pain is increasing in many countries [17,18], there is still limited research on related prescribing patterns [19]. Moreover, it is not clear whether some patterns are associated with a longer duration of opioid use. Different studies suggest that this condition might be associated with a higher risk of drug abuse and dependence [20,21,22]. Indeed, while treatment of acute pain is rarely associated with the development of opioid abuse/dependence, chronic opioid therapy may result in opioid abuse/dependence in 3% to 19% of patients [22]. Results of a large Norwegian study suggest that even if only one-fourth of patients starting opioid therapy for chronic nonmalignant pain enter long-term treatment, a large proportion of this minority develops or is at risk for developing problematic opioid use and addiction [23].

The paper is structured as follows. Section 2 introduces the prescription data that will be used in the analysis. Section 3 presents the core steps of the SSA approach (data coding Section 3.1, measurement of dissimilarities between sequences Section 3.2, the application of clustering methods to the dissimilarity matrix Section 3.3). Section 4 and Section 5 introduce complexity measures and tools for graphical visualization. Section 6 shows how to include clusters obtained by SSA into predictive models. The main results are reported in Section 7 and Section 8, while Section 9 concludes the article with some final remarks and indications for future research.

## 2. Data Source and Cohort Identification

Italian administrative data were used to show the application of SSA to drug prescription data. Italy has a tax-based, universal coverage National Health System (NHS) organized in three levels: national, regional (21 regions), and local (on average 10 Local Health Authorities (LHA) per region). Healthcare is managed to the inhabitants by the LHA according to their regular address. The local health authority databases contain, among others, annual drug prescriptions dispensed by local pharmacies. In this study, we used data from the LHA of Novara (about 350,000 residents), in the Piedmont region. For each prescription contained in the database, the following information was retrieved: patient ID, age and sex, dispensing date, ATC code, formulation, number of packages and Defined Daily Dose (DDD). For the purposes of this study, we focused on opioid prescriptions (ATC code N02A), which were classified in two groups [24], “strong” (S) and “weak” (W). The former group included morphine, hydromorphone, oxycodone, fentanyl, buprenorphine and tapentadol, while the latter included codeine and tramadol. For each prescription, we assumed that the consumption started on the day of dispensing. The duration of each prescription was calculated, dividing the total amount of active substance dispensed by the relevant DDD, which is assumed to represent the average maintenance dose per day for a drug used for its main indication in adults (https://www.whocc.no/atc_ddd_index/, accessed on 1 January 2020).

New users of opioid therapy for chronic non-cancer pain were identified in the LHA according to the following inclusion criteria:Started an opioid treatment between 1 January 2012 and 31 November 2012. One year of look-back was applied to exclude prevalent users;Had at least two prescriptions of opioids, with the second one occurring within 70 days after the first one;Have been treated with opioids for at least one year (i.e., the last opioid prescription was dispensed at least one year after the first one);Did not have any hospital discharge record in 2011 and 2012 with a diagnosis of cancer.

Cancer patients were not considered in this study for two reasons: (i) the different time pattern in the opioid use when compared with patients with chronic or non-neoplastic diseases; (ii) the use of painkillers often until the very end of their life, nullifying the possibility of studying time to cessation of the therapy.

## 3. The SSA Approach

In SSA, the prescription history of a subject can be described as a sequence of different categorical states. In our case, states are defined from the type of opioid therapy (strong/weak/none) assumed during each period of time. For instance, a subject could start taking weak opioids for two weeks, then have a 12-week break and finally start strong opioids until the end of the study. SSA allows to compare sequences among the different subjects with respect to the succession of their component states and to identify common patterns. In general, the core SSA process can be broken down into three main steps: data coding, measurement of dissimilarities between sequences, and application of a clustering method to identify sequence patterns [25].

### 3.1. Data Coding

Medication data are usually retrieved in SPELL format, in which each row represents a single prescription. To carry out SSA, it is necessary first to convert these data into an STS (state sequence) format, where each row represents a patient, as shown in Table 1.

The following step specifies the alphabet of the sequence. This is a discrete list of all possible states appearing in the data. In our case, the alphabet is composed of the following states: S, W, P, representing, respectively, strong opioid prescription, weak opioid prescription and pause period, during which no opioid prescription is dispensed. For example, the sequence SSWWPWP represents the following pattern: strong, strong, weak, weak, pause, weak, pause. Once the alphabet has been specified, it is necessary to define the length of the study period and the time unit (e.g., day, week, month, etc.) of the analysis, also called sequence granularity. These two aspects will determine the start, the end, and, therefore, the length of the sequences to be analyzed. In our example, we set the time unit equal to one week. This choice was due to the fact that opioid prescriptions typically cover one week of therapy in most of patients.

We classified the exposure status on each subject in the different weeks according to the most frequent state that appeared in the considered time unit. Thus, for example, if in a specific week a subject had three days of strong opioid therapy and four of weak opioid therapy, the state for that time unit was W. In the case of equally frequent states, we selected the first state that appears in the sequence. For each subject, the sequence started with the first prescription of opioids dispensed in 2012 and ended one year after the index prescription. Thus, for each subject, the sequence was constituted of 52 states. Notably, we included only subjects that had at least one year of opioid therapy. This means that we did not have to deal with right-truncated data (due, for example, to either cessation or death). Moreover, as the study period started with the time of the first prescription, all the sequences were left-aligned. Thus, we did not have to deal with missing states at the beginning of the sequence for some subjects (due to delayed entry in the cohort). The absence of missing states in our database made the analysis simpler and overcome the risk of creating an artifactual cluster of patients with missing states [25].

### 3.2. Measurement of Dissimilarities between Sequences

Once sequences were created, we estimated the degree of dissimilarity between each pair of sequences. To this aim, different measures of dissimilarity are available. Optimal Matching (OM) is one of the most used methods in bioinformatics for the evaluation of DNA sequences. The basic idea behind OM is to measure the dissimilarity of two sequences by considering how much effort is required to transform one sequence into the other one, applying some basic edit-operations (insertion, deletion, substitution) [9]. However, in settings different from bioinformatics (such as social sciences and, arguably, pharmacoepidemiology), these edit-operations do not have a direct interpretation. This makes it difficult to obtain meaningful results in these cases. For this reason, the longest common subsequence (LCS) metric was introduced as a special case of the OM to be applied to social sciences [26]. Briefly, the longer a subsequence that can be shared among two different sequences, the more they are considered similar. This metric allows to construct a dissimilarity matrix containing all the pairwise distances among the sequences (subjects) included in the dataset.

### 3.3. Clustering Methods

The dissimilarity matrix was then used to cluster sequences and to identify different prescription patterns [27]. Cluster analysis identifies a set of multivariate methods designed to select and group homogeneous patients with respect to their quantitative or qualitative characteristics (distances). In this study, we used hierarchical agglomerative cluster methods, which are based on an iterative procedure to assemble observations in groups [28]. These methods start considering every subject as a single group and, step by step, end with a unique cluster composed of the whole set of observed subjects. In each step, an observation is associated with an already existing cluster or forms a new cluster based on the smallest observed distance. Among the different hierarchical methods, we adopted the Ward method [29], which simultaneously minimizes the within-cluster variance and maximizes the among-clusters variance at each iteration step. The optimal number of clusters can be chosen either theoretically or empirically. In our case, the optimal number of groups was determined empirically by visual inspection of the dendrogram, a graphical representation of the cluster hierarchy based on the distance among the groups identified in the iteration (Appendix A).

## 4. Complexity Measures for Sequence Patterns

The distance between sequences is useful for representing their dissimilarities but does not describe other important features, such as the number of states, the overall time spent in each state, and the number of transitions between states. On the other side, these three statistics may be difficult to interpret on their own, also considering they are strictly correlated to each other. To overcome this limit, different indicators have been proposed to summarize longitudinal characteristics of individual sequences [30]. In particular, we considered longitudinal Shannon’s entropy and Elzinga’s Turbulence. Longitudinal Shannon’s Entropy is a weighted average of the time spent in each state within the same sequence (in our case, the total amount of time spent under a specific drug regimen). Elzinga’s Turbulence [31] is instead a composite measure considering the number of possible subsequences and the variance of the consecutive time spent in each state. Further theoretical details on these indicators can be found in [30].

## 5. Graphical Visualization of Sequences

In SSA, graphical representations of patterns play an important role [30] in the communication of the results. In particular, the index plot is composed of horizontal lines for each sequence, separating the states with different colors [32]. When it is directly used on a large number of data, as in our case, it is usually not very helpful as it fails to capture specific patterns and leads to the over-plotting phenomenon [32]. However, this plot becomes very useful when it is employed after clustering methods, as it allows to identify characteristic features of the different clusters. The state distribution plot is another available graphical tool. It displays the general pattern of the whole set of data [30], showing the proportion of patients in the different states at each time point. Jumps and peculiar moves among vertical distributions can be interpreted following [33,34,35].

## 6. Predictive Models

Finally, results of sequence mining techniques can be integrated with classical statistical methods to evaluate the association between specific prescription patterns and health-related outcomes. Here, we showed an application of this approach using the results of cluster analysis and complexity measures to predict the future discontinuation of opioid therapy. This is an important outcome in opioid chronic therapy, as a longer duration of use is associated with a higher risk of abuse and dependence on these medications. For the sake of this analysis, subjects were followed from the beginning of the second year of therapy until 31 November 2015 or death, whatever occurred first, to evaluate discontinuation. Clusters identified by SSA during the first year of therapy and groups based on the tertiles of the distribution of complexity measures were used as independent variables to predict subsequent treatment discontinuation in a time-to-event analysis. Specifically, the association between clusters and tertile subgroups with the outcome was evaluated through the Kaplan–Meier method and multivariable Cox regression models adjusted by age and gender.

Preliminary data management and manipulation of the dataset was carried out using SAS. All the SSA techniques used in this paper were performed using the R package TraMineR.

## 7. SSA Results

Based on our selection criteria, 469 new users of opioids for the treatment of chronic non-cancer pain were identified and included in the analysis. Their characteristics are summarized in Table 2. The majority of these patients were women (71%), with a mean age of 72 years. In general, weak opioids were used more than strong ones. During the first year, 79% of subjects received weak opioid prescriptions vs. 50% receiving strong opioids. Notably, a substantial proportion of subjects (29%) received both. Women tended to use both weak and strong opioids more than men. However, use of opioids was generally low during the first year (on average, about 9 weeks of use of any type of opioids in both sexes) (Table 2). This suggests that most of the subjects had intermittent therapy during the first period of treatment. The state distribution plot, reported in Figure 1, shows changes in the proportion of patients treated with weak and strong opioids during the first year of treatment. Almost 80% of subjects started with weak opioids, but the “pause” state soon became the dominant one (because of the intermittent regimen). Moreover, since the first weeks, the number of users of weak and strong opioids became much more similar and remained constant over time. This suggests that many subjects had an early escalation from weak to strong opioids.

Results of the cluster analysis are reported in Table 3 and Figure 2. Six main clusters were identified, with cluster 1, 2 and 5 corresponding to patients mainly using weak opioids and accounting for 70% of the cohort (Table 3). By contrast, clusters 3, 4 and 6 included mainly strong opioid users. Cluster 1 and 4 had the largest use of weak and strong opioids, respectively, while clusters 2 and 3 identified subjects with the lowest use of any type of opioids.

## 8. Prediction of Treatment Discontinuation

The results of the cluster analysis were used to predict the future discontinuation of opioid therapy in the cohort. Figure 3 shows that the probability of treatment discontinuation was different among the clusters (*p* < 0.001). These results were confirmed in the Cox analysis reported in Table 4. Compared to cluster 2 (the reference group), subjects belonging to cluster 6 had a 63% lower probability of stopping opioid treatment (adjusted hazard ratio 0.36; 95% confidence interval: 0.20 to 0.65). The other clusters displayed an intermediate behavior, with their curves falling between those of cluster 2 and 6.

The probability of treatment discontinuation has also been studied according to complexity measures (Table 4 and Figure 4). Regarding entropy, the probability of discontinuing the therapy decreased throughout the tertile groups (*p* < 0.001). Compared to the first tertile, patients belonging to the third tertile had a 57% lower probability of discontinuation (adjusted hazard ratio 0.36; 95% confidence interval: 0.30 to 0.64). Similar findings were observed using turbulence (*p* < 0.001).

## 9. Further Considerations

Using real-world data, in this paper, we showed how SSA allows the identification of patterns in prescriptions that might not be detected using standard statistical approaches and how these patterns are associated with future discontinuation of opioid therapy. In recent years, epidemiological literature showed an increasing interest in different data mining techniques to characterize drug consumption [36,37]. Multivariate exploratory methods, such as principal component analysis, multiple correspondence analysis and agglomerative hierarchical clustering. Refs. [36,37] have been proposed, as well as latent class models [4,36,38,39]. While all these approaches have their advantages, they usually work on a limited number of class-defining variables and are thus not very suitable to evaluate complex longitudinal patterns of prescriptions. Moreover, as their implementation to real data is quite complex, their practical application in pharmacoepidemiology has been limited so far. To this extent, SSA could constitute a simpler alternative to evaluate prescription data. To date, only few epidemiological studies have exploited the potential of SSA, mainly to evaluate patterns of healthcare utilization of patients with different conditions [25,40,41,42]. Geographical disparities have been identified in care consumption on a cohort of pregnant women [40] and on the elderly with end-stage renal disease [41] with SSA. The latter has been recently employed also to detect the association between ethnicity and socio-economic status on children’s body mass index trajectories [43].

To the best of our knowledge, the authors of [14] were the first to apply SSA in pharmacoepidemiology to investigate patterns of respiratory drugs treatments. Nonetheless, the use of SSA in their paper was indeed rather limited, as it only employed index plot as a visualization tool and compared it with other methodologies. Two very recent studies made a more comprehensive use of SSA with several advantages. On the one hand, SSA was used to assess the effectiveness of regulatory restrictions in “sleeping pill” prescriptions, detecting a reduced exposure in subgroups of long-term users [15]. On the other hand, SSA helped to provide an overall characterization of disease-modifying therapies in patients with multiple sclerosis both at individual and geographical levels [16].

There are some methodological aspects that should be kept in mind when performing SSA of pharmacoepidemiological data. First, the strategy used for the assignation of the exposure status (the state) should be appropriate for the research question. In the case of equally frequent states within a specific time unit, we selected the first state that appears in the sequence. However, in some situations it could be more sensible to consider an extra state allowing for multiple exposures. For example, in our case-study, this would have resulted in an additional “SW” state. Second, the effect of the chosen time-granularity on results should be carefully evaluated. This issue was already raised by Vanasse et al. in the context of healthcare utilization [42], and it could also be relevant when SSA is applied to analyze prescriptions of specific medications. Thus, before drawing any inference from SSA of pharmacoepidemiologic data, we recommend carrying out sensitivity analyses using different levels of time-granularity.

Third, in some situations, it could be useful to analyze the patterns of different types of drugs at the same time. In this regard, multichannel sequence analysis allows several thematic sequences for one patient to be studied simultaneously and could represent in the future a very interesting extension of standard SSA for pharmacoepidemiological studies [25].

Given the nature of EHR data, SSA tools represents a potentially insightful integration of traditional approaches to study drug exposure and adverse effects. The method can be applied within the traditional case-control [44] and cohort designs [45] but also within the more recent self-matched designs [46], i.e., the self-controlled case series [47], case-crossover [48], and sequence symmetry analysis [49]. Typically, those designs are implemented using parametric regression models and carry important statistical properties [47,48], but they also require several assumptions to be met and yield measures that only answer well-defined research questions. Conversely, SSA aims at producing exploratory outputs throughout the construction of both visual and numeric data summaries that can also be easily integrated into the most widely used regression models [15,16]. SSA might also represent a valuable practical tool in hypothesis-free signal detection in pharmacovigilance [46,50] as it accommodates the analysis of a large dataset, increasingly present, for instance, in national administrative databases.

Research contributions of this study are mainly twofold. First, this paper provides an educational primer of the most important learning concepts and methods of SSA. The latter has been applied to a pharmacoepidemiology working example to propose a guide/tutorial to the scientific community. Second, we have highlighted the need of a shift in perspective to prescription data analysis from a cross-sectional and “compartmentalized” approach to a holistic one, which enables extensive exploitation of the information available throughout the application of a variety of data mining tools readily available in main statistical software.

Our working example has a limited sample size, no updated information or follow-up, and limited patient information. For this reason, it should be not considered to directly inform clinical practice but only for illustrative purposes. Further research might address these issues conducting a thorough epidemiological analysis to shed light on opioid use patterns at national, regional, and province levels. We also believe that extensive investigations would be required to assess how SSA, in pharmacoepidemiology, performs better at providing critical insights when compared to traditional approaches.

Finally, the SSA methodology could be combined with other novel approaches, such as pharmacodynamic-based classification of drugs, built on the capacity of single medications to interact with specific receptors [51,52]. In some cases, this strategy could be useful for providing a pharmacological interpretation of the results of SSA.

## 10. Conclusions

Our contribution shows the potential of the SSA method in pharmacoepidemiologic studies. This technique is easy to use, and its intrinsic visual nature may help investigators to untangle the latent information within prescription datasets, facilitating the individuation of specific patterns and possible inappropriate use of medications. In turn, this may also help to evaluate the effect of drug policies and adherence to medical guidelines. All these features make SSA a promising tool for future pharmacoepidemiologic research.

## Figures and Tables

**Figure 1 ijerph-18-13398-f001:**
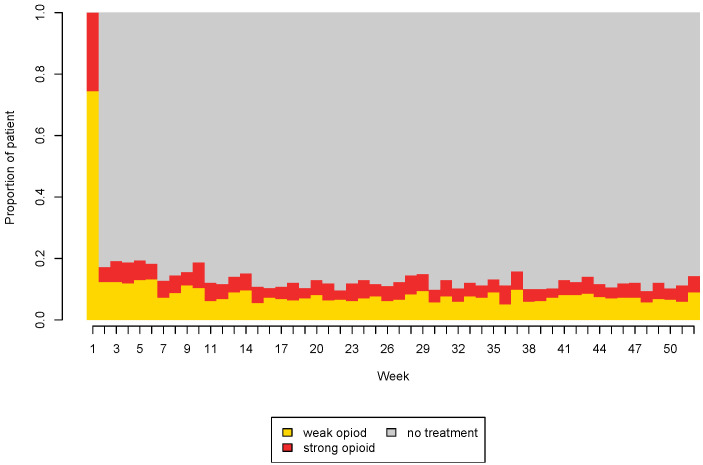
State distribution plot. Evolution of the proportion of patients using different types of opioids during the first year of therapy.

**Figure 2 ijerph-18-13398-f002:**
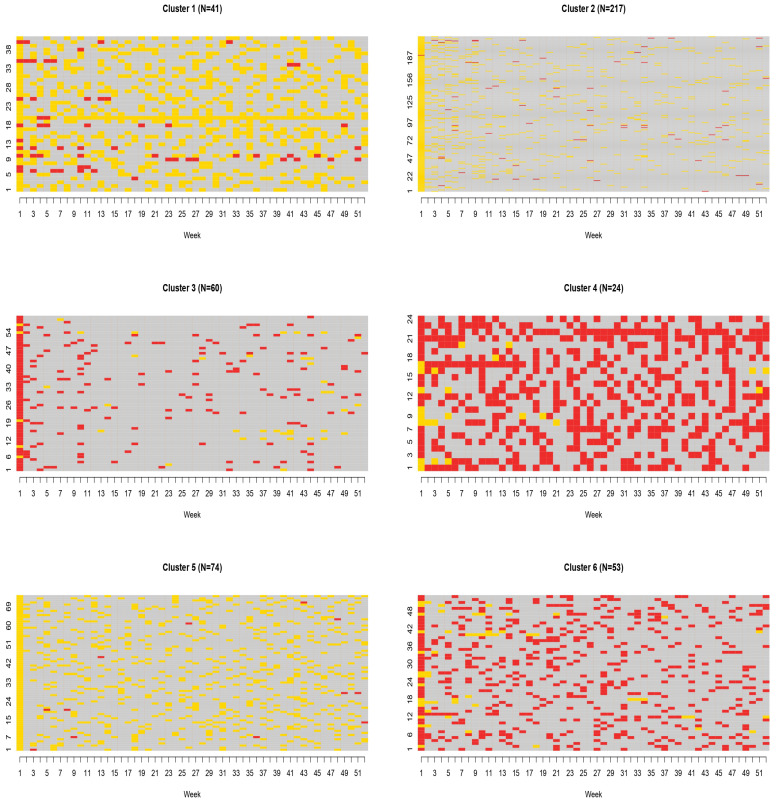
Index plots for weekly regimen use during the first year of opioid therapy.

**Figure 3 ijerph-18-13398-f003:**
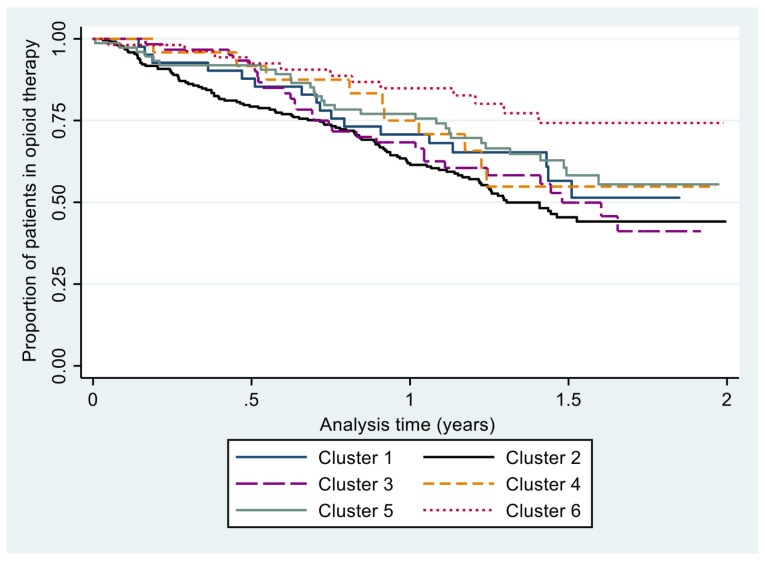
Time to discontinuation of the opioid therapy in the different clusters.

**Figure 4 ijerph-18-13398-f004:**
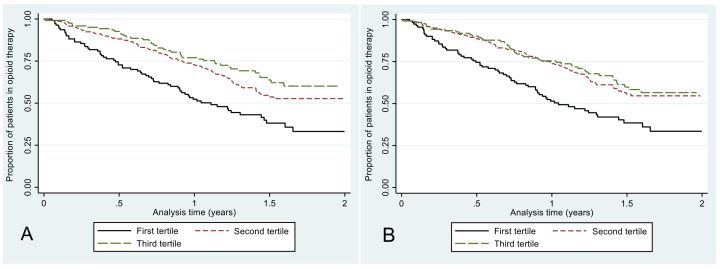
Time to discontinuation of the opioid therapy according to different tertiles of entropy (**A**) and turbulence (**B**).

**Table 1 ijerph-18-13398-t001:** Example of State Sequence (STS) format. W (in yellow) and S (in red) stand for weak and strong opioid medication, respectively. P (in gray) stands for pause (no treatment).

ID Patient	Week 1	Week 2	Week 3	Week 4	Week 5	...	Week 52
1	**W**	**W**	**W**	**W**	**P**	**...**	**P**
2	**W**	**P**	**S**	**S**	**W**	**...**	**S**
3	**S**	**S**	**S**	**S**	**P**	**...**	**P**

**Table 2 ijerph-18-13398-t002:** Characteristics of the sample.

	Men	Women
N	139	330
Age (SD)	66.9 (14.6)	73.3 (12.9)
Weeks with weak opioids in first year (SD)	4.5 (5.4)	5.0 (5.0)
Weeks with strong opioids in first year (SD)	3.1 (5.0)	3.3 (5.1)
Number of prescriptions in first year (SD)	8.2 (6.5)	9.6 (6.1)
Discontinued by the end of the FU (%)	62 (44.6%)	143 (43.3%)

**Table 3 ijerph-18-13398-t003:** Characteristics of the subjects by clusters.

	Cluster 1	Cluster 2	Cluster 3	Cluster 4	Cluster 5	Cluster 6
N	41	217	60	24	74	53
Women (%)	33 (80.5%)	142 (65.4%)	42 (74%)	18 (70%)	55 (75%)	40 (75.5%)
Age (SD)	75.7 (11.4)	70.0 (13.9)	72.4 (11.8)	65.4 (18.1)	74.7 (13.4)	70.6 (13.2)
Weeks with weak opioids in first year (SD)	16.2 (6.7)	3.7 (1.7)	0.7 (1.4)	1 (1.5)	9.2 (1.8)	1.1 (1.7)
Weeks with strong opioids in first year (SD)	1.5 (2.6)	1.3 (0.7)	3.4 (1.9)	19.8 (5.7)	0.2 (0.5)	9.3 (2.7)
Number of prescriptions in first year (SD)	17.9 (8.7)	5.3 (2.9)	7.3 (4.1)	20.5 (5.0)	11.0 (2.7)	12.7 (4.3)
Discontinued by the end of the FU (%)	17 (41%)	108 (50%)	29 (48%)	10 (42%)	29 (39%)	12 (23%)

**Table 4 ijerph-18-13398-t004:** Association of clustering and complexity measures with time to discontinuation of opioid therapy. Results from crude and adjusted Cox regression. * Results adjusted by age and sex.

		Crude	Adjusted *
	Variable	HR (95% CI)	HR (95% CI)
Clusters	Cluster 1	0.74 (0.44–1.23)	0.75 (0.45–1.26)
	Cluster 2	1 (ref)	1 (ref)
	Cluster 3	0.89 (0.59–1.33)	0.90 (0.59–1.35)
	Cluster 4	0.70 (0.37–1.34)	0.70 (0.36–1.34)
	Cluster 5	0.65 (0.43–0.98)	0.66 (0.44–1.00
	Cluster 6	0.36 (0.20–0.65)	0.36 (0.30–0.65)
Entropy	1st tertile	1 (ref)	1 (ref)
	2nd tertile	0.58 (0.41–0.78)	0.56 (0.41–0.77)
	3rd tertile	0.43 (0.30–0.64)	0.44 (0.30–0.64)
Turbulence	1st tertile	1 (ref)	1 (ref)
	2nd tertile	0.53 (0.39–0.73)	0.53 (0.39–0.73)
	3rd tertile	0.49 (0.33–0.71)	0.49 (0.34–0.72)

## Data Availability

The commented R code and the dataset are provided as online Appendix A.

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
