# Peer review of "Use of State Sequence Analysis in Pharmacoepidemiology: A Tutorial"

_ijerph, 2021, doi:10.3390/ijerph182413398_

Round 1
Reviewer 1 Report
Comment to authors:
I have gone through this paper thoroughly.
The authors introduce an adcanved method (i.e., state sequence analysis (SSA)) to handle the medical issue.
The content of this paper is very easy to follow.
The experimental design and result of the model is well-defined and reliable.
One of my concern is the authors need to identify why this model performs better than the previous related methods.
What is the main contribution of this paper.
Please add more managerial implication in this study (that is, add one section to describe the managerial implication)
I think this paper is interesting and has considerable poblication potential.
Author Response
First of all, we would like to sincerely thank the reviewers for their valuable time and precious comments, which definitely helped us to improve the paper.
Below are our point-by-point answers to the reviewers, in blue font.
Reviewer 1
I have gone through this paper thoroughly. The authors introduce an advanced method (i.e., state sequence analysis (SSA)) to handle the medical issue. The content of this paper is very easy to follow. The experimental design and result of the model is well-defined and reliable.
We thank the reviewer for this positive remark which really encourage us to work on the revision of our manuscript.
One of my concern is the authors need to identify why this model performs better than the previous related methods.
Thank you for this comment. We have added in the paper the following text. “Given the nature of EHR data, SSA tools represents a potentially insightful integration of traditional approaches to study drug exposure and adverse effects. The method can be applied within the traditional case- control [Stolley,1992] and cohort designs [Creagh,1992] but also within the more recent self- matched designs [Hallas,2014], i.e. the self-controlled case series [Whitaker,2009], case-crossover [Consiglio,2013], sequence symmetry analysis [Lai,2017]. Typically, those designs are implemented using parametric regression models and carry important statistical properties [Whitaker,2009; Consiglio,2013] but they also require several assumptions to be met and yield measures that only answer well-defined research questions. Conversely, SSA aims at producing exploratory outputs throughout construction of both visual and numeric data summaries that can also be easily integrated into the most widely used regression models [Istvan, 2021; [LeBlanc, 2021]]. SSA might also represent a valuable practical tool in hypothesis-free signal detection in pharmacovigilance [Bate, 2019; Hallas, 2018] as it accommodates the analysis of large dataset, increasingly present for instance, in national administrative database.” (Section 9, lines 303 - 315)
We have also included in the limitations of this study the need for further investigations to assess the performance of the tools compared to using classical approaches: “We believe, also, that extensive investigations would be required to assess how SSA, in pharmacoepidemiology, performs better in providing critical insights when compared to traditional approaches.” (Section 9, lines 327 - 330)
What is the main contribution of this paper.
We have better highlighted our contributions in the manuscript adding the following text. “Research contributions of this study are mainly twofold. First, this paper provides an educational primer of the most important learning concepts and methods of SSA. The latter has been applied on a working example in pharmacoepidemiology to propose a guide/tutorial to the scientific community. Second, we highlight the need of a shift in perspective to prescription data analysis from a cross-sectional and “compartmentalized” approach to a holistic one, which enables extensive exploitation of the information available throughout the application of a variety of data mining tools readily available in main statistical software” (Section 9, lines 316 - 323)
Please add more managerial implication in this study (that is, add one section to describe the managerial implication)
Thank you to have highlighted this aspect. There are many managerial implications of our contributions that we summarize in what follows: “[...] SSA aims at producing exploratory outputs throughout construction of both visual and numeric data summaries that can also be easily integrated into the most widely used regression models [Istvan, 2021; LeBlanc, 2021]. SSA might also represent a valuable practical tool in hypothesis-free signal detection in pharmacovigilance [Bate, 2019; Hallas, 2018] as it accommodates the analysis of large dataset, increasingly present for instance, in national administrative database.” (Section 9, lines 310-315)
I think this paper is interesting and has considerable publication potential.
Thank you for your encouraging remarks. We look forward to hearing from you regarding our submission. We would be glad to respond to any further questions and comments that the reviewer may have.

Reviewer 2 Report
This paper provides an educational primer of the most important learning concepts and methods of SSA, including measurement of dissimilarities between sequences, the application of clustering methods to identify sequence patterns, the use of complexity measures for sequence patterns, the graphical visualization of sequences, and the use of SSA in predictive models.
Please divide the "Conclusion" section independently (i. e., 10. Conclusions)
Author Response
Reviewer 2
This paper provides an educational primer of the most important learning concepts and methods of SSA, including measurement of dissimilarities between sequences, the application of clustering methods to identify sequence patterns, the use of complexity measures for sequence patterns, the graphical visualization of sequences, and the use of SSA in predictive models. Please divide the "Conclusion" section independently (i. e., 10. Conclusions)
Thank you for your suggestion. We applied this change in the manuscript as recommended.

Reviewer 3 Report
There are six key problems in your article:
- There is no detailed literature search and analysis justifying your use of SSA.
- Use of an outdated database form 2012
- A small sample (139 males, 330 females)
- You excluded cancer patients who are important opioid users. What are your reasons for excluding them?
- The patients are absent. You provide no information about their main illnesses, comorbidities, or Charlson frailty index, or outcomes for these patients.
- What use can clinicians make of your findings?
Because your study lacks a detailed literature search on SSA, and analysis of the risks of bias and outcomes from SSA analysis that added new findings compared to previous literature please perform this literature search and report on it.
“So far, in pharmacoepidemiology these techniques have been mainly used to identify adverse drug reactions [3]”
[Please provide details. You argue that SSA is key to your analysis so please provide examples showing its effectiveness in previous studies].
“While SSA has been long used in social sciences, especially for labour and family mobility research [7], its use in pharmacoepidemiology is still in its infancy [8]. ”
[Again please provide details and examples showing its effectiveness].
“Moreover, is not clear whether some patterns are associated with a longer duration of opioid use, which in turn constitutes an important risk for developing drug abuse and dependence [12].”
[Please provide more evidence for this important assertion and search for systematic reviews and provide their summaries]
“started an opioid treatment between 01/01/2012 and 31/12/2012. One year of look-back was applied to exclude prevalent users; 2. had at least two prescriptions of opioids, with the second one occurring within 70 days after the first one; 3. have been treated with opioids for at least one year (i.e. the last opioid prescription was dispensed at least one year after the first one); 4. did not have any hospital discharge record in 2011 and 2012 with a diagnosis of cancer.”
[Please provide the reason why you are using a database that is 11 years old. Have you followed the patients up to now? If not can you update your file to a recent period – there have been more initiatives to limit opioid prescriptions in recent years in several countries].
“only few epidemiological studies have exploited the potential of SSA mainly to evaluate patterns of healthcare utilization of patients with different conditions.”
[please comment in detail what these studies found and what SSA contributed that was not known using other methods].
Please provide conclusions relevant to clinical practice and improved patient care.
Author Response
Reviewer 3
There are six key problems in your article:
1. There is no detailed literature search and analysis justifying your use of SSA.
Thank you to point out this lack. We elaborated the discussion accordingly. “To fill this gap, in this paper we show the application of state sequence analysis (SSA) to pharmacoepidemiological data to evaluate the temporal order of prescriptions and to identify latent complex patterns. Without assuming priori hypothesis, SSA is an effective tool to study distinctive features of homogeneous groups of sequences, exploiting their pairwise dissimilarities in unsupervised clustering [Studer, 2011]. The SAA indeed looks at the life sequence as a single unit of analysis, extracts fundamental descriptive information and makes the data easier to comprehend, differently from event history analysis [Billari, 2005]. Identifying also sequence typical trajectories and recurring structures, it is considered the most suitable tool for signal detection in healthcare databases [Arnaud, 2017].
While SSA has been long used in social sciences, especially for labour and family mobility research [Billari, 2001; Demont, 2008; Bras, 2010; Buergin, 2017; Ritschard, 2018], its use in pharmacoepidemiology is still in its infancy [Parkin, 2018; Itsvan, 2021; LeBlanc, 2021]. SSA allows a simple and compact representation, identical to the one used to code DNA molecules, of life courses [Billari, 2001]. Thus, in social sciences, SSA is employed, for instance, to model demographic projections based on microsimulation methods [Billari, 2001], family file events [Buergin, 2017], early employment insecurity [Ritschard, 2018], pathways to adulthood [Bras, 2010], and the prevalence of nuclear families [Demont, 2008]. Moreover, SSA in pharmacoepidemiology proves to be effective in evaluating the impact of regulatory measure on the prescription of sedative medications [Itsvan, 2021]; describing longitudinal patterns of disease-modifying therapies usage grouping population [LeBlanc, 2021; and investigating the conformity of prescribing practices of respiratory drugs treatments [Parkin, 2018].” (Section 1, lines 33 - 51)
-
Use of an outdated database form 2012
Our contribution is not directly oriented on the study of a particular pharmacoepidemiological phenomenon and does not aim to draw any inference from this specific set of data. Rather, it is just a tutorial to stimulate the use of SSA in pharmacoepidemiology. From this point of view, the fact that the dataset is old is, in our opinion, not particularly relevant. In any case, we accommodated your comment enriching our contribution with a discussion of its limitations, among which the outdated database “Our working example has a limited sample size, not updated information or follow-up, and patients' information are limited. For this reason, it should be not considered to directly inform clinical practice, but only for illustrative purposes. Further research might address these issues conducting a throughout epidemiological analysis to shed some light on opioids’ use patterns at national, regional, and province levels” (Section 9, lines 324-328) -
A small sample (139 males, 330 females)
Please see response to point 2. As we do not aim to draw any inference from this specific set of data but we are just exploiting it for illustrative purposes, the size of the sample is not of vital importance. However, we also add this aspect on the limitations “Our working example has a limited sample size, not updated information or follow-up, and patients' information are limited. For this reason, it should be not considered to directly inform clinical practice, but only for illustrative purposes.” (Section 9, lines 324 -326)
-
You excluded cancer patients who are important opioid users. What are your reasons for excluding them?
We excluded cancer patients because the time pattern of use of opioid is very different from that of patients with chronic, non-neoplastic diseases. For this reason, epidemiological studies on the use of these drugs usually do not mix these two types of patients (see for example Fredheim 2013, Edlund 2010 and Martin 2011, reported in the references of our paper). In particular, cancer patients often use painkillers until the very end of their life, while we were interested in studying whether some patterns of use were associated with time to cessation of the therapy. We added a sentence in the methods section to explain it in a clearer way. “Cancer patients were not considered in this study for two reasons: i) the different time pattern in the opioid use when compared with patients with chronic or non- neoplastic diseases; ii) the use of painkillers often until the very end of their life, nullifying the possibility of studying time to cessation of the therapy.” (Section 2, lines 102- 105)
5. The patients are absent. You provide no information about their main illnesses, comorbidities, or Charlson frailty index, or outcomes for these patients.
See response to point 2, where we pointed out the illustrative purposes of these data and the fact that we do not draw any inference from them. To accommodate your comment, we have added this aspect on the limitations. “Our working example has a limited sample size, not updated information or follow-up, and patients' information are limited. For this reason, it should be not considered to directly inform clinical practice, but only for illustrative purposes. Further research might address these issues to conduct a throughout epidemiological analysis to shed some light on opioids’ use patterns at national, regional, and province levels.” (Section 9, lines 324-328)
What use can clinicians make of your findings?
This paper is not directly aimed at supporting clinicians in their practical choices, but rather to researchers who use pharmacoepidemiologic data. For this reason, we submitted our paper to a journal of public health with a strong methodological flavor and not to a clinical journal. Indeed, in this paper we provided a tutorial to show how to use a new analytic method. As we pointed out in the text of the paper, our results should not be used to draw any specific inference on these drugs but are provided only for illustrative purposes. We add in the manuscript what follows “Our working example has a limited sample size, not updated information or follow-up, and patients' information are limited. For this reason, it should be not considered to directly inform clinical practice, but only for illustrative purposes.” (Section 9, lines 324-326)
Because your study lacks a detailed literature search on SSA, and analysis of the risks of bias and outcomes from SSA analysis that added new findings compared to previous literature please perform this literature search and report on it.
Thank you to have highlighted this lack. Following response to point 1, we have added further details and references.
“To fill this gap, in this paper we show the application of state sequence analysis (SSA) to pharmacoepidemiological data to evaluate the temporal order of prescriptions and to identify latent complex patterns. Without assuming priori hypothesis, SSA is an effective tool to study distinctive features of homogeneous groups of sequences, exploiting their pairwise dissimilarities in unsupervised clustering [Studer, 2011]. The SAA indeed looks at the life sequence as a single unit of analysis, extracts fundamental descriptive information and makes the data easier to comprehend, differently from event history analysis [Billari, 2005]. Identifying also sequence typical trajectories and recurring structures, it is considered the most suitable tool for signal detection in healthcare databases [Arnaud, 2017].
While SSA has been long used in social sciences, especially for labour and family mobility research [Billari, 2001; Demont, 2008; Bras, 2010; Buergin, 2017; Ritschard, 2018], its use in pharmacoepidemiology is still in its infancy [Parkin, 2018; Itsvan, 2021; LeBlanc, 2021]. SSA allows a simple and compact representation, identical to the one used to code DNA molecules, of life courses [Billari, 2001]. Thus, in social sciences, SSA is employed, for instance, to model demographic projections based on microsimulation methods [Billari, 2001], family file events [Buergin, 2017], early employment insecurity [Ritschard, 2018], pathways to adulthood [Bras, 2010], and the prevalence of nuclear families [Demont, 2008]. Moreover, SSA in pharmacoepidemiology proves to be effective in evaluating the impact of regulatory measure on the prescription of sedative medications [Itsvan, 2021]; describing longitudinal patterns of disease- modifying therapies usage grouping population LeBlanc, 2021; and investigating the conformity of prescribing practices of respiratory drugs treatments [Parkin, 2018].” (Section 1, lines 31-51)
“Geographical disparities have been identified in care consumption on a cohort of pregnant women [Le Meur,2015] and on elderly with end-stage renal disease [Le Meur,2019] with SSA. The latter has been recently employed also to detect the association between ethnicity and socio- economic status on children body mass index trajectories [Moreno-Black,2016]. [...]On the other hand, SAA has provided an overall characterization of disease-modifying therapies in patients with multiple sclerosis both at individual and geographical level [LeBlanc, 2021].” (Section 9, lines 273 - 276)
“So far, in pharmacoepidemiology these techniques have been mainly used to identify adverse drug reactions [3]” [Please provide details. You argue that SSA is key to your analysis so please provide examples showing its effectiveness in previous studies].
We have integrated the literature review as follow.
“So far, in pharmacoepidemiology these techniques have been mainly used to identify adverse drug reactions [Arnaud, 2017], while their application to the analysis of drug prescriptions to identify longitudinal use patterns is still limited [Franklin 2013].[...] Without assuming priori hypothesis, SSA is an effective tool to study distinctive features of homogeneous groups of sequences, exploiting their pairwise dissimilarities in unsupervised clustering [Studer, 2011]. The SAA indeed looks at the life sequence as a single unit of analysis, extracts fundamental descriptive information and makes the data easier to comprehend, differently from event history analysis [Billari, 2005]. Identifying also sequence typical trajectories and recurring structures, it is considered the most suitable tool for signal detection in healthcare databases [Arnaud, 2017].” (Section 1, lines 24 - 40)
Moreover: “Geographical disparities have been identified in care consumption on a cohort of pregnant women [Le Meur,2015] and on elderly with end-stage renal disease [Le Meur,2019] with SSA. The latter has been recently employed also to detect the association between ethnicity and socio-economic status on children body mass index trajectories [Moreno-Black,2016].
At the best of our knowledge, [Parkin, 2018] were the first to apply SSA in pharmacoepidemiology, to investigate patterns of respiratory drugs treatments. Nonetheless, the use of SSA in their paper was indeed rather limited, as it only employed index plot as a visualization tool and compared it with other methodologies. Two very recent studies made a more comprehensive use of SSA with several advantages. On the one hand, SAA assessed effectiveness of regulatory restrictions in “sleeping pill” prescriptions detecting a reduced exposure in subgroups of long-term users [Istvan, 2021]. On the other hand, SAA has provided an overall characterization of disease-modifying therapies in patients with multiple sclerosis both at individual and geographical level [LeBlanc, 2021].” (Section 9, lines 273 - 285)
“While SSA has been long used in social sciences, especially for labour and family mobility research [7], its use in pharmacoepidemiology is still in its infancy [8]. ”[Again please provide details and examples showing its effectiveness].
According to your comment we have added further references to enhance the effectiveness of the SSA use. “While SSA has been long used in social sciences, especially for labour and family mobility research [Billari, 2001; Demont,2008; Bras,2010; Buergin,2017, Ritschard,2018], its use in pharmacoepidemiology is still in its infancy [Parkin, 2018; Itsvan,2021; LeBlanc,2021]. SSA allows a simple and compact representation, identical to the one used to code DNA molecules, of life courses [Billari, 2001]. Thus, in social sciences, SSA is employed, for instance, to model demographic projections based on microsimulation methods [Billari, 2001], family file events [Buergin, 2017], early employment insecurity [Ritschard, 2018], pathways to adulthood [Bras, 2010], and the prevalence of nuclear families [Demont, 2008]. Moreover, SSA in pharmacoepidemiology proves to be effective in evaluating the impact of regulatory measure on the prescription of sedative medications [Itsvan, 2021]; describing longitudinal patterns of disease- modifying therapies usage grouping population LeBlanc, 2021; and investigating the conformity of prescribing practices of respiratory drugs treatments [Parkin, 2018].” (Section 1, lines 33-51)
“Moreover, is not clear whether some patterns are associated with a longer duration of opioid use, which in turn constitutes an important risk for developing drug abuse and dependence [12].” [Please provide more evidence for this important assertion and search for systematic reviews and provide their summaries]
We expanded the paragraph accordingly to the reviewer’s request:
“Moreover, is not clear whether some patterns are associated with a longer duration of opioid use. Different studies suggest that this condition might be associated with a higher risk of drug abuse and dependence [Martin 2010, Edlund 2010, Edlund 2010b]. Indeed, while treatment of acute pain is rarely associated with development of opioid abuse/dependence, chronic opioid therapy may result in opioid abuse/ dependence in 3% to 19% of patients [Edlund 2010]. Results of a large Norwegian study suggest that that even only one-fourth of patients starting opioid therapy for chronic nonmalignant pain enter long-term treatment, a large proportion of this minority develops or is at risk for developing problematic opioid use and addiction [Fredheim 2013].” (Section 1, lines 56-64)
“started an opioid treatment between 01/01/2012 and 31/12/2012. One year of look-back was applied to exclude prevalent users; 2. had at least two prescriptions of opioids, with the second one occurring within 70 days after the first one; 3. have been treated with opioids for at least one year
(i.e. the last opioid prescription was dispensed at least one year after the first one); 4. did not have any hospital discharge record in 2011 and 2012 with a diagnosis of cancer.” [Please provide the reason why you are using a database that is 11 years old. Have you followed the patients up to now? If not can you update your file to a recent period – there have been more initiatives to limit opioid prescriptions in recent years in several countries].
Please refer to point 2, regarding the illustrative purposes of the dataset and the methodological aim of the present paper. To accommodate your comment, we have added the following text in the limitations of this study to open also over further research. “Our working example has a limited sample size, not updated information or follow-up, and patients' information are limited. For this reason, it should be not considered to directly inform clinical practice, but only for illustrative purposes. Further research might address these issues to conduct a throughout epidemiological analysis to shed some light on opioids’ use patterns at national, regional, and province levels.” (Section 9, lines 324- 328)
“only few epidemiological studies have exploited the potential of SSA mainly to evaluate patterns of healthcare utilization of patients with different conditions.”[please comment in detail what these studies found and what SSA contributed that was not known using other methods].
We have enriched the discussion with further references and with the following text.
“Geographical disparities have been identified in care consumption on a cohort of pregnant women [Le Meur,2015] and on elderly with end-stage renal disease [Le Meur,2019] with SSA. The latter has been recently employed also to detect the association between ethnicity and socio- economic status on children body mass index trajectories [Moreno-Black, 2016].” (Section 9, lines 273-279)
“On the other hand, SAA has provided an overall characterization of disease-modifying therapies in patients with multiple sclerosis both at individual and geographical level [LeBlanc, 2021].” (Section 9, lines 281-285)
Please provide conclusions relevant to clinical practice and improved patient care.
As we already stated, our study is not directly aimed to clinicians and should not considered providing conclusions directly relevant to clinical practice. Instead, we expanded our conclusions to make clear why we think that this study is relevant to the pharmacoepidemiological practice. “Given the nature of EHR data, SSA tools represents a potentially insightful integration of traditional approaches to study drug exposure and adverse effects. The method can be applied within the traditional case-control [Stolley,1992] and cohort designs [Creagh,1992] but also within the more recent self-matched designs [Hallas,2014], i.e. the self-controlled case series [Whitaker,2009], case-crossover [Consiglio,2013], sequence symmetry analysis [Lai,2017] . Typically, those designs are implemented using parametric regression models and carry important statistical properties [Whitaker,2009; Consiglio,2013] but they also require several assumptions to be met and yield measures that only answer well-defined research questions. Conversely, SSA aims at producing exploratory outputs throughout construction of both visual and numeric data summaries that can also be easily integrated into the most widely used regression models [Istvan, 2021; LeBlanc, 2021]. SSA might also represent a valuable practical tool in hypothesis-free signal detection in pharmacovigilance [Bate, 2019; Hallas, 2018] as it accommodates the analysis of large dataset, increasingly present for instance, in national administrative database.
Research contributions of this study are mainly twofold. First, this paper provides an educational primer of the most important learning concepts and methods of SSA. The latter has been applied on a pharmacoepidemiology working example to propose a guide/tutorial to the scientific community. Second, we have highlighted the need of a shift in perspective to prescription data analysis from a cross-sectional and “compartmentalized” approach to a holistic one, which enables extensive exploitation of the information available throughout the application of a variety of data mining tools readily available in main statistical software.
Our working example has a limited sample size, not updated information or follow-up, and patients' information are limited. For this reason, it should be not considered to directly inform clinical practice, but only for illustrative purposes. Further research might address these issues to conduct a throughout epidemiological analysis to shed some light on opioids’ use patterns at national, regional, and province levels.” (Section 9, lines 303-328)

Round 2
Reviewer 3 Report
Thanks to the authors, who have comprehensively replied to the reviewers' suggestions.